# Hypoperfusion Index Ratio as a Surrogate of Collateral Scoring on CT Angiogram in Large Vessel Stroke

**DOI:** 10.3390/jcm10061296

**Published:** 2021-03-21

**Authors:** Chun-Min Wang, Yu-Ming Chang, Pi-Shan Sung, Chih-Hung Chen

**Affiliations:** Department of Neurology, National Cheng Kung University Hospital, College of Medicine, National Cheng Kung University, Tainan 704, Taiwan; ro2003yes@gmail.com (C.-M.W.); lchih@mail.ncku.edu.tw (C.-H.C.)

**Keywords:** hypoperfusion index ratio, collateral circulation, collateral scoring, CTA, CTP, large vessel occlusion

## Abstract

Background: This study was to evaluate the correlation of the hypoperfusion intensity ratio (HIR) with the collateral score from multiphase computed tomography angiography (mCTA) among patients with large vessel stroke. Method: From February 2019 to May 2020, we retrospectively reviewed the patients with large vessel strokes (intracranial carotid artery or proximal middle cerebral artery occlusion). HIR was defined as a Tmax > 10 s lesion volume divided by a Tmax > 6 s lesion volume, which was calculated by automatic software (Syngo.via, Siemens). The correlation between the HIR and mCTA score was evaluated by Pearson’s correlation. The cutoff value predicting the mCTA score was evaluated by receiver operating characteristic analysis. Result: Ninety-four patients were enrolled in the final analysis. The patients with good collaterals had a smaller core volume (37.3 ± 24.7 vs. 116.5 ± 70 mL, *p* < 0.001) and lower HIR (0.51 ± 0.2 vs. 0.73 ± 0.13, *p* < 0.001) than those with poor collaterals. A higher HIR was correlated with a poorer collateral score by Pearson’s correlation. (r = −0.64, *p* < 0.001). The receiver operating characteristic (ROC) analysis suggested that the best HIR value for predicting a good collateral score was 0.68 (area under curve: 0.82). Conclusion: HIR is a good surrogate of collateral circulation in patients with acute large artery occlusion.

## 1. Introduction

Evaluating pial (leptomeningeal) collateral status is of great importance in predicting the evolution of infarction [1], predicting the prognosis of acute ischemic stroke [2], and selecting eligible patients for endovascular thrombectomy (EVT) [3]. Leptomeningeal collateral flow can be assessed by conventional angiography, computed tomography angiography (CTA) (including single-phase CTA, dynamic CTA, and multiphase CTA) and magnetic resonance imaging (MRI) [4]. Different scoring systems have been proposed, and most of them are semiquantitative measures [5]. In clinical practice, multiphase CTA (mCTA) has become one of the most reliable and rapid techniques to visualize collateral circulation [6]. However, there may be potentially an interrater difference in reading the result of mCTA and obtaining collateral scores on mCTA in real-world setting.

Along with mCTA, computed tomography perfusion (CTP) plays a role in decision making regarding the management of acute stroke, especially before EVT [7]. The development of automatic postprocessing software for CTP gives physicians more quantitative and rapid measures to evaluate the infarct core and potentially salvageable tissue. Calculated by automatic software, the hypoperfusion intensity ratio (HIR) was defined as the Tmax > 10 s lesion volume (Tmax10) divided by the Tmax > 6 s lesion volume (Tmax6) [8]. The HIR has been shown to predict the rate of infarct growth and functional outcome at 90 days after stroke in the DEFUSE 2 cohort; thus, it is thought to be a clinical parameter that evaluates the degree of collateral circulation [8]. In another retrospective study, patients who met the American Heart Association guidelines for thrombectomy were more likely to have a lower HIR [9].

A recent study showed that HIR was correlated with collateral circulation in digital subtraction angiography (DSA), suggesting a cutoff value (HIR < 0.4) as the best prediction for good DSA collaterals [10]. There may be a correlation between collateral scores on mCTA and HIR, but the cutoff value of HIR for the prediction of good collaterals on mCTA may be different from the cutoff value to predict good DSA collaterals.

On the other hand, the studies above all used RAPID software (iSchemaView, Menlo Park, CA, USA) as postprocessing software. Although other software programs have been developed and have shown some degree of agreement with RAPID [11,12], no study has demonstrated that the HIR calculated by other automatic software correlates with collateral status.

In this study, we aimed to establish the association between the HIR calculated by Syngo.via and the collateral score by mCTA and determine the best cutoff value for predicting good collaterals on mCTA.

## 2. Materials and Methods

### 2.1. Patient Inclusion, Population, and Clinical Data

National Cheng Kung University Hospital (NCKUH) is a 1320-bed tertiary medical center in southern Taiwan that can provide intravenous tissue plasminogen activator injection (IV-tPA) and endovascular thrombectomy (EVT). Between 800 and 900 patients with acute ischemic stroke are admitted to our stroke ward annually. As a participating hospital of the nationwide Taiwan Stroke Registry (TSR) [13], NCKUH has been maintaining prospective stroke registries according to the TSR protocol since 2006. Our comprehensive stroke center prospectively enrolls patients who present to our hospital within 10 days after stroke onset and receive CT and/or MRI for the index stroke. The patients’ demographic characteristics and medical history were recorded according to a predefined system.

In this study, we retrospectively identified consecutive patients receiving CTA and CTP scans on arrival at our emergency department for acute stroke management between February 2019 and May 2020. By assessing each patient’s CTA data, we enrolled patients with occlusions in either the internal carotid artery (ICA) or the M1 and M2 branches of the middle cerebral artery (MCA). Patients without large vessel occlusion (LVO) or with occlusion in the anterior cerebral artery (ACA), posterior cerebral artery (PCA), posterior circulation, or multiple sites were excluded because the collateral scores were non applicable. The demographic data, last known well time or onset time, initial National Institutes of Health Stroke Scale (NIHSS) score, comorbidities, details of IV-tPA and/or EVT, and modified Rankin scale (mRS) at discharge, were obtained from our registry system.

All patients needed to complete written consents prior to receive brain imaging. This study was approved by the Institutional Review Board of National Cheng Kung University Hospital (B-ER-109192).

### 2.2. Multiphase Computed Tomography Angiography Collateral Score, Hypoperfusion Intensity Ratio, and the Eligibility of EVT

The mCTA protocol was described in a previous study [6]. In brief, three phases (peak arterial phase, peak venous phase, and late venous phase) of consecutive scanning with an interval of 8 s were obtained, allowing for time-resolved assessment. The mCTA collateral scores (range from 0 to 5) are defined as follows: Grade 5—no filling delay compared to the asymptomatic contralateral hemisphere, normal pial vessels in the affected hemisphere; Grade 4—a filling delay of one phase in the affected hemisphere, but the extent and prominence of pial vessels is the same; Grade 3—a filling delay of two phases in the affected hemisphere, or a delay of one phase with a significantly reduced number of vessels in the ischemic territory; Grade 2—a filling delay of two phases in the affected hemisphere with a significantly reduced number of vessels in the ischemic territory, or one phase delay showing regions without visible vessels; Grade 1—only a few vessels are visible in the affected hemisphere in any phase; Grade 0—no vessels visible in the affected hemisphere in any phase.

The mCTA collateral scores were independently assessed by two raters (Wang C-M and Chang Y-M). Those results with different mCTA collateral scores were further discussed at the research conference by these two raters. A final mCTA collateral score was given after discussion and agreement. An mCTA collateral score of 3 or lower indicates poor collateral status [14].

The CTP images were postprocessed by the software Syngo.via CT Neuro Perfusion (version VB30 HF03; Siemens Healthcare, Erlangen, Germany). Tmax is defined as the time to maximum of the residue function obtained by deconvolution [15]. The volume of the ischemic core, penumbra and perfusion mismatch were automatically calculated based on cerebral blood flow (<30%) and Tmax (>6 s) lesion volume. The HIR was defined as the Tmax > 10 s lesion volume divided by the Tmax > 6 s lesion volume.

The eligibility criteria of EVT at our site are mainly based on the guidelines from the American Heart Association/American Stroke Association (AHA/ASA) [16] and Taiwan Stroke Society [17]. In brief, for patients with LVO within 6 h of stroke onset and an Alberta stroke program early CT score (ASPECTS) ≥ 6, EVT was considered unless patients had poor baseline conditions, such as a pre-mRS score greater than 2, terminal cancer status, unstable vital signs or multiple comorbidities. Perfusion imaging may provide complementary information alongside CTA for neurointerventionists, especially for patients within 6 h to 24 h of last known normal. For those patients, EVT was considered case by case based on the criteria of the DAWN [18] or DEFUSE 3 trial [19].

### 2.3. Outcome and Statistical Analysis

The continuous variables (age, NIHSS score, time after stroke onset, HIR, mCTA collateral score, ischemic core, penumbra, and perfusion mismatch volume) are expressed as the means ± standard deviation (SD) or median, quartiles and interquartile range (IQR). The nominal variables (medical history of comorbidity and medication, occlusion sites, IV-tPA and EVT) were summarized as frequency descriptive analyses. The interrater reliability of mCTA score was measured by using Cohen’s kappa coefficient.

The subjects were divided into subgroups based on the mCTA collateral score (good collaterals (scores 4–5) vs. poor collaterals (scores 0–3)). Univariate analyses were performed to compare the age, initial stroke severity assessed by initial NIHSS score, ischemic core volume, penumbra volume, perfusion mismatch volume and perfusion ratio between groups by using independent T-test or Mann–Whitney U-test; the sex, comorbidity, medication history, treatment with IV-tPA or EVT, and post-EVT TICI score between groups by using Pearson’s chi-squared test. The correlation between the HIR and mCTA collateral score was calculated using Pearson’s correlation. Statistical tests are considered significant at a < 0.05 level. Receiver operating characteristic (ROC) curve analysis was performed to determine an HIR threshold for predicting good collaterals, which was defined as mCTA collateral scores of 4–5.

## 3. Results

From February 2019 to May 2020, 341 patients with acute ischemic stroke underwent CTA and CTP at NCKUH. After excluding those without LVO (*n* = 153), those with stroke in the posterior circulation and PCA territory (*n* = 56), those in the ACA territory (*n* = 9), those with bilateral or multiple occlusion sites (*n* = 3) and those with poor image quality (e.g., failure to be processed by software, poorly enhanced vessels, severe motion artifacts, etc.) or missing data (*n* = 26), 94 patients were enrolled in the final analysis (male/female: *n* = 59/36) (Figure 1). The mean age was 72 (SD: 12.9, range: 30–94), and the median NIHSS score was 21 (IQR: 14–27). The occlusion sites were at the ICA (*n* = 23), M1 (*n* = 42) and M2 (*n* = 29). The median HIR was 0.65 (IQR: 0.47–0.74), and the median mCTA score was 4 (IQR: 2–4). The mCTA score showed substantial agreement between the two raters with a kappa value of 0.64.

There were no significant differences in age, sex, history of hypertension, diabetes mellitus, coronary artery disease, congestive heart failure, prior antiplatelet or anticoagulant use, or tobacco use between patients with good and poor collaterals (Table 1). The patients with good collaterals had significantly lower stroke severity (median NIHSS = 14, IQR: 10–21) than those with poor collaterals (median NIHSS = 25, IQR: 21–30, *p* < 0.001). There were also more patients with good collaterals receiving IV-tPA (44.2% versus 16.7%, *p* = 0.004). There were no significant differences in the percentage of patients receiving EVT between the two groups.

The patients with good collaterals had smaller cores (37.3 ± 24.7 vs. 116.5 ± 70 mL, *p* < 0.001) and Tmax6 (120 ± 64.9 vs. 203.5 ± 88 mL, *p* < 0.001) and Tmax10 volumes (59.2 ± 31.1 vs. 152 ± 82.6 mL, *p* < 0.001) and lower HIRs (0.51 ± 0.2 vs. 0.73 ± 0.13, *p* < 0.001), as well as lower mRS scores (2 (IQR: 1–3.75) vs. 5 (IQR: 2–6), *p* = 0.02) at discharge (Table 2).

A higher HIR was correlated with a poorer collateral score by Pearson’s correlation (r = −0.64, *p* < 0.001) (Figure 2). The ROC analysis suggested that the best value for predicting a good collateral score was 0.68, with a sensitivity of 0.76, specificity of 0.81, and area under the curve of 0.82 (Figure 3).

## 4. Discussion

Our study found that the HIR is correlated with the mCTA collateral score in patients with acute occlusions at the ICA, M1, or M2 segment of the MCA, with 0.68 being the best value that predicts a good collateral score by Syngo.via CT Neuro Perfusion software.

In clinical practice, we may evaluate collateral status directly via CTA, and the collateral scores were correlated with clinical outcomes, even with reperfusion by IV-tPA and EVT [2,20,21]. However, there are some pitfalls in scoring the collateral status from CTA. It is somehow subjective and rater-dependent, and the raters need to be trained to reduce the interrater variability. Objective automatic software may be helpful in the clinical scenario of managing acute ischemic stroke. HIR, automatically calculated by software, is defined as Tmax10 divided by Tmax6. Tmax6 was shown to predict penumbra well in a previous study [22], while Tmax10 was found to represent the most endangered tissue with extremely delayed perfusion [23]. HIR may be considered a quantitative measure of collateral blood flow to the brain as “tissue-level collaterals”. One previous study demonstrated that HIR was correlated with collateral circulation in DSA in patients during EVT [10]. Another analysis of the SWIFT PRIME study also showed that collateral status was correlated with relative blood volume and HIR by using RAPID software [24]. Our findings confirm the concept of using HIR as an indicator for collateral circulation even with a different software package and provide a particular cutoff value of HIR to predict good collateral status by using Syngo.via software.

In the study mentioned above [8], the ROC analysis showed that an HIR > 0.4 had a sensitivity of 0.66 and a specificity of 0.70 for predicting poor collateral flow. Another study [10] also revealed that an HIR < 0.403 best predicted good angiographic collaterals with a sensitivity of 0.79 and specificity of 0.56. Both reports used RAPID software. We found that the cutoff value for predicting good collaterals in mCTA was 0.68 by using Syngo.via software. This may result from the different algorithms of image processing and chosen parameters in different software packages. A study reported that the infarct core predicted by Syngo.via will meet good agreement with RAPID if changing the parameters to CBV < 1.2 mL/100 mL and applying an additional smoothing filter [12], while another study suggested that the predicted volume of the infarct core calculated by Syngo.via could be concordant with RAPID if the relative cerebral blood flow threshold is changed to <20% [25]. In the same study, when analyzed as a subgroup, in patients with LVOs (ICA and M1 occlusion), there was no statistically significant difference between the calculated values for the core and hypoperfusion volumes. To the best of our knowledge, there have been no studies comparing and correlating the Tmax10 or HIR of the two software packages. Further study is warranted to correlate the HIR acquired by different software programs. Despite some difference in predicted volume of core and penumbra, Syngo.via and RAPID software showed high concordance in correctly triaging patients into “go or no-go” for EVT in real-world settings [26].

Potential delay may exist between activating the thrombectomy team and the actual reperfusion time. Better collateral circulation may extend the survival period of the penumbra. A study showed that patients with good collaterals had a smaller infarct core and higher mismatch ratio in ICA and M1/M2 occlusion and within 12 h of stroke onset [27]. Interestingly, in the DEFUSE 3 cohort, good collaterals were associated with reduced ischemic core growth but not neurologic outcome in the late therapeutic window [28]. Another study also showed that in patients with LVO who underwent endovascular intervention, collateral status was strongly associated with MCA territory final infarct volumes but not correlated with favorable outcomes at discharge [24]. The authors’ explanation was that good collaterals preserved the watershed area of the ACA/MCA and MCA/PCA but had no influence on certain critical brain regions (such as the precentral gyrus and the posterior limb of the internal capsule), which have larger impacts on functional independence. On the other hand, in the DEFUSE 2 cohort, final infarction volume increased in association with HIR quartiles as well as infarct growth regardless of reperfusion. After adjusting for the factor of early reperfusion, a favorable outcome was still associated with a low HIR [8]. Therefore, by using HIR as a surrogate for “tissue-level collateral assessment”, a stroke neurologist may have additional information to accelerate the patient selection of EVT and predict the outcome. If our result is further validated in other independent database, multiphase CTA may not be necessary in most of the cases because the HIR could already represent the collateral status and even correlate prognosis better. Patients could benefit from reduced exposure of radiation and contrast medium, and we may save more time during pre-EVT evaluation.

There were some limitations of our study. First, this was a retrospective observational study in a single medical center, and sampling bias was inevitable, although we collected consecutive acute stroke patients who received CTA/CTP. Second, despite the specific definition, the mCTA score is a subjective scoring system that may have interrater differences, and there are different scoring systems for collaterals in CTA [29], which are not fully comparable. Third, we only included patients with acute stroke and large vessel occlusion patients. Those with occlusion sites other than the ICA or MCA were excluded; thus, the correlation of the mCTA score and HIR may not be reliable at other occlusion sites or in post-acute stage of stroke. Fourth, the proportion of patients undergoing EVT was comparable between good and poor collaterals in our study. This may be due to the higher proportion of M2 occlusion in patients with good collaterals, and EVT in patients with M2 occlusions is optional at our sites. Fifth, the sensitivity and specificity of this threshold by Syngo.via has not been validated in other independent database. Furthermore, the chronic stenosis of large vessels, old stroke and poor cardiac output tremendously affect the image quality of CTP. Therefore, the above conditions may interfere with the relationship between collateral status and HIR. Last but not least, this study was performed in an Eastern Asian population; thus, extrapolating our findings to other ethnicities must be done with caution.

## 5. Conclusions

In our study, we found that a lower HIR correlates with a good mCTA collateral core in patients with occlusions in the ICA and M1 and M2 segments of the MCA. The best cutoff value of HIR is 0.68 to predict good collaterals by Syngo.via. The HIR is a good surrogate of tissue-level collateral status, even in different automatic software packages. However, the best HIR cutoff value to predict good collaterals may be adjusted by different software programs.

## Figures and Tables

**Figure 1 jcm-10-01296-f001:**
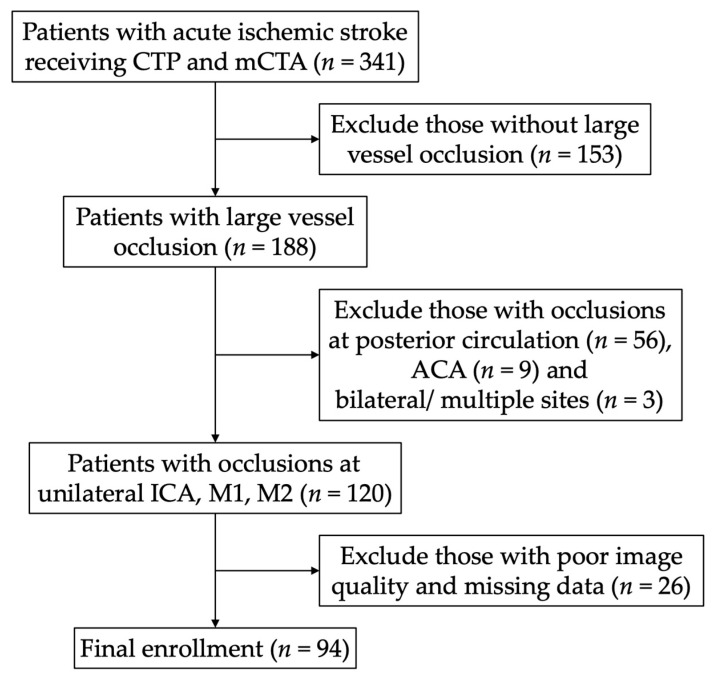
The flow chart of enrollment. CTP, computed topography perfusion; mCTA, multiphase computed topographic angiography; ACA anterior cerebral artery; ICA, internal carotid artery; M1/M2, M1, and M2 segments of the middle cerebral artery.

**Figure 2 jcm-10-01296-f002:**
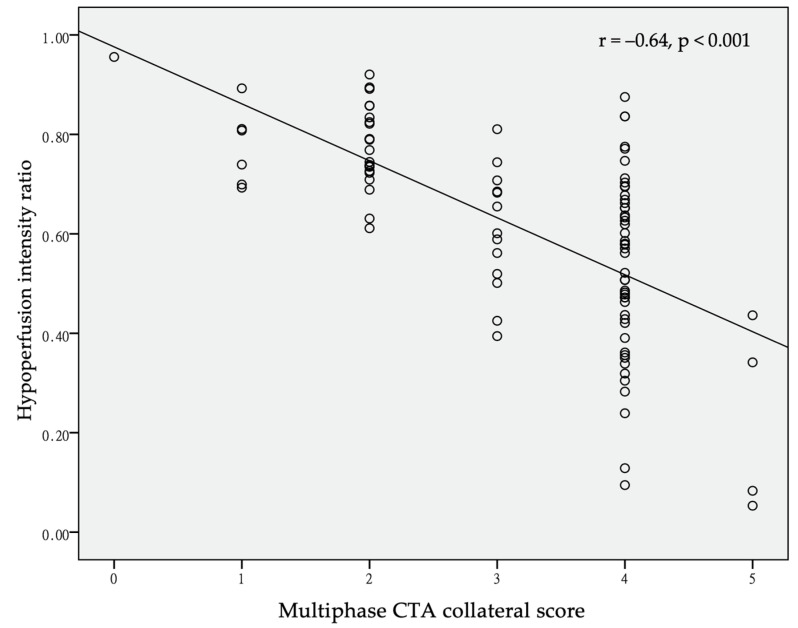
Scatter plot of hypoperfusion index (HIR) and multiphase CT angiography (mCTA) collateral score. Pearson’s r = –0.64, *p* < 0.001.

**Figure 3 jcm-10-01296-f003:**
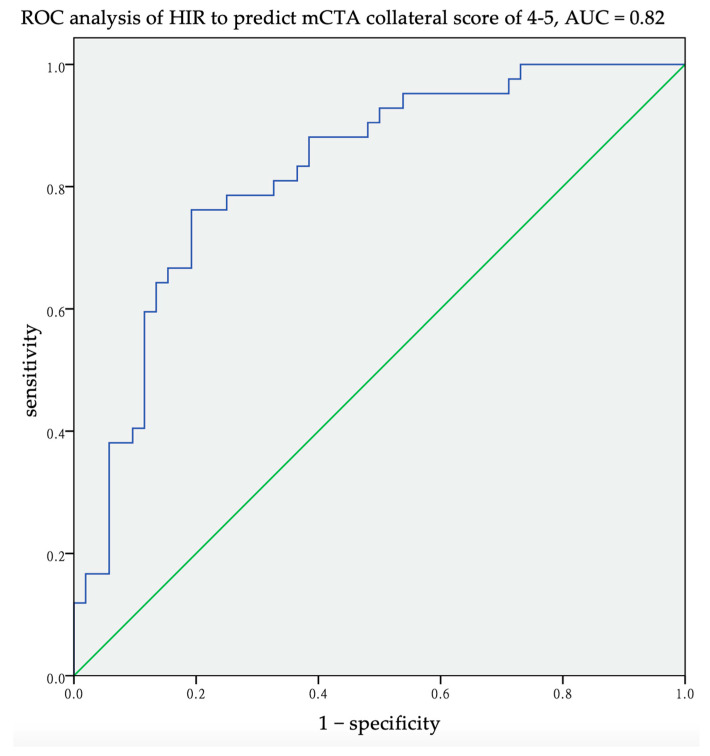
Receiver operating characteristic (ROC) analysis of the hypoperfusion index (HIR) to predict good collateral by multiphase CT angiography (mCTA) collateral score (4 or 5). The best predicted value of HIR was 0.68, with a sensitivity of 76%, specificity of 81% and area under curve (AUC) of 0.82.

**Table 1 jcm-10-01296-t001:** Demographic characteristics of patients with poor collaterals (score 0–3) and good collaterals (score 4–5) based on multiphase CT angiography collateral score in acute ischemic stroke.

Characteristics	All (*n* = 94)	Poor Collaterals (*n* = 42)	Good Collaterals (*n* = 52)	*p* Value
Age (years) (mean (SD))	71.9 (12.9)	73.0 (11.3)	71.0 (14.1)	0.439
Male	55 (58.5%)	23 (54.8%)	32 (61.5%)	0.507
Medical history				
Hypertension	71 (75.5%)	32 (76.2%)	39 (75%)	0.894
Diabetes Mellitus	35 (37.2%)	15 (35.7%)	20 (38.5%)	0.784
Hyperlipidemia	64 (68.1%)	25 (59.5%)	39 (75%)	0.110
Atrial fibrillation	42 (44.7%)	24 (57.1%)	18 (34.6%)	0.029
Coronary artery disease	19 (20.2%)	12 (28.6%)	7 (13.5%)	0.070
Congestive heart failure	10 (10.6%)	7 (16.7%)	3 (5.8%)	0.088
Prior AP	25 (26.9%)	11 (26.2%)	14 (26.9%)	0.936
Prior AC	13 (13.8%)	7 (16.7%)	6 (11.5%)	0.474
Smoker	24 (25.8%)	7 (16.7%)	17 (33.3%)	0.095
Initial NIHSS (median (IQR))	20.5 (14–27)	25 (21–30)	14 (10–21)	<0.001
Lesion site				0.021
ICA	23 (24.5%)	11 (26.2%)	12 (23.1%)	
M1	42 (44.7%)	24 (57.1%)	18 (34.6%)	
M2	29 (30.9%)	7 (16.7%)	22 (42.3%)	
Initial SBP (mmHg) (median (IQR))	152.5 (137.75–171.25)	154 (137–180)	151.5(137.5–167.75)	0.451
IV-tPA	30 (31.9%)	7 (16.7%)	23 (44.2%)	0.004
EVT	32 (34.0%)	12 (28.6%)	20 (38.5%)	0.314
Onset to ER (median (IQR))	154 (61–311.75)	172.5 (123–277.75)	111.50 (37.5–364.25)	0.050
Onset to CTP (median (IQR))	40.50 (23.75–75.25)	30.50 (20–49.5)	52 (31.50–82.75)	0.005

NIHSS, National Institutes of Health Stroke Scale; ICA, internal carotid artery; M1/M2, M1, and M2 segments of the middle cerebral artery; EVT, endovascular thrombectomy; CTP, computed tomography perfusion.

**Table 2 jcm-10-01296-t002:** Core volume, Tmax > 6- and 10-s lesion volume, hypoperfusion index ratio (HIR) and modified Rankin scale (mRS) at discharge in patients with poor collaterals (score 0–3) and good collaterals (score 4–5) based on the multiphase CT angiography collateral score.

Characteristics	All (*n* = 94)	Poor Collaterals (*n* = 42)	Good Collaterals (*n* = 52)	*p* Value
Core volume (mL) (SD)	72.7 (63.7)	116.5 (70.0)	37.3 (24.7)	<0.001
Tmax > 6 volume (mL) (SD)	157.3 (86.4)	203.5 (88.0)	120.0 (64.9)	<0.001
Tmax > 10 volume (mL) (SD)	100.7 (75.2)	152.0 (82.6)	59.2 (31.1)	<0.001
HIR (SD)	0.61 (0.20)	0.73 (0.13)	0.51 (0.20)	<0.001
Discharge mRS (IQR)	4 (1.25–5)	5 (2–6)	2 (1–3.75)	0.021

HIR, hypoperfusion intensity ratio.

## Data Availability

The data was available upon reasonable email request.

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
