# Peer review of "Hypoperfusion Index Ratio as a Surrogate of Collateral Scoring on CT Angiogram in Large Vessel Stroke"

_jcm, 2021, doi:10.3390/jcm10061296_

Round 1
Reviewer 1 Report
This a well written manuscript that evaluates the association between the HIR and the collateral score by mCTA and tries to determine the best cutoff value for predicting good collaterals on mCTA .
I would like to make some comments in order to improve your manuscript.
- There is a strong correlation betwwen HIR and collateral score but this is met only in patients with specific characteristics , that are described in the inclusion criteria. this might be added to the study limitations.
- you could explain more entensively, if there is a clinical impact from this correlation and whether the patients management is altered by this obsernation.
- a scatter plot could help the reader better to assess the correlation of the values (colateral score-HIR)
Author Response
Thanks for the reviewer's detailed comments. We had made point-by-point reply and made revisions.
Reviewer 1
- There is a strong correlation between HIR and collateral score, but this is met only in patients with specific characteristics, that are described in the inclusion criteria. this might be added to the study limitations.
Reply:
Thanks for the reviewer’s comments. We had revised our limitation according to reviewer’s comments, and put more emphasis on the characteristics of our included patients, including in the settings of "acute” stroke and “large vessel occlusion”
Revision in the limitation, line 267-270:
Third, we only included patients with acute stroke and large vessel occlusion patients. Those with occlusion sites other than the ICA or MCA were excluded; thus, the corre-lation of the mCTA score and HIR may not be reliable at other occlusion sites or in post-acute stage of stroke.
- You could explain more extensively, if there is a clinical impact from this correlation and whether the patient’s management is altered by this observation.
Reply:
Thanks for the reviewer’s comments. We had added more possible clinical impact of our findings in the discussion, and we tried not to exaggerate because we still thought this concept need to be validated in larger independent database.
Revision in the discussion, line 257-261:
If our result is further validated in other independent database, multiphase CTA may not be necessary in most of the cases because the HIR could already represent the collateral status and even correlate prognosis better. Patients could benefit from reduced exposure of radiation and contrast medium, and we may save more time during pre-EVT evaluation.
- A scatter plot could help the reader better to assess the correlation of the values (collateral score-HIR)
Reply:
Thanks for the reviewer’s recommendation. We had added the scatter plot as the figure 2. Thanks for the reviewer’s comments.
Reviewer 2 Report
The authors investigated in this study the relation between the hypoperfusion intensity ratio HIR on computed tomography perfusion (CTP) measured by Syngovia and the collateral score as visualized on mCTA. Also, they provided a cut-off HIR value to predict the status of the collaterals (good or poor).
Studying the relationship between the HIR and collaterals on DSA has been previously investigated. However, in the present study another software package is used. It is relevant to evaluate whether this can be done using this software package as well. In addition, determining the cut-off value using another software is another relevant aim. The manuscript is well written,
However, I have a few comments:
Major comments
Major comment:
- The study is very similar to that of Xia et al, Clinical Radiology, Volume 74, Issue 12, December 2019, Pages 956-961, but this paper is not referred to.
- A limitation of the study is that the sensitivity and specificity with this threshold are not validated in an independent dataset.
Minor comments
- Page 1. line 41-44. A reference should be provided here.
- Page 1, Line 36: ‘However, there is an interrater difference in reading the result of mCTA and o ijoy K. Menon et al 2015 btaining collateral scores on mCTA’. The authors have used the same protocol as in reference 6, where an excellent interrater reliability was reported: “Interrater reliability for pial arterial filling with multiphase CT angiography was excellent (n= 30, κ = 0.81, P < .001).”
- Almost same information was provided in lines (71-74) and (81-84), so these paragraphs could be merged.
- Line 79-80: “Patients without large vessel occlusion (LVO) or with occlusion in the anterior cerebral artery (ACA), posterior cerebral artery (PCA), posterior circulation, or multiple sites were excluded.” Can the authors explain why these patients were excluded?
- Line 90. Although the CTA protocol has been described a previous study, it would be helpful if the main CTA parameters would be described, including the timing of the different phases with respect to contrast injection.
- Page 1 and abstract: Tmax should be defined
- In the statistical analysis paragraph they didn’t mention that they calculate inter-rater agreement using the kappa test.
- A reference is missing for lines 107-109.
- Twenty-six patients were excluded because of poor image quality and missing data. How the image quality assessed? Did they use an image quality score?
- Figure 1. There seems to be a calculation mistake in Figure 1. The differences do not match number of patients that were excluded.
- Spelling mistakes in Table 1: years; Table 2: Volume (twice)
Author Response
Thanks a lot for the reviewer’s detailed comments and suggestions. We had made point-by-point reply and made revisions.
Reviewer 2
The authors investigated in this study the relation between the hypoperfusion intensity ratio HIR on computed tomography perfusion (CTP) measured by Syngovia and the collateral score as visualized on mCTA. Also, they provided a cut-off HIR value to predict the status of the collaterals (good or poor).
Studying the relationship between the HIR and collaterals on DSA has been previously investigated. However, in the present study another software package is used. It is relevant to evaluate whether this can be done using this software package as well. In addition, determining the cut-off value using another software is another relevant aim. The manuscript is well written,
However, I have a few comments:
Major comments
- The study is very similar to that of Xia et al, Clinical Radiology, Volume 74, Issue 12, December 2019, Pages 956-961, but this paper is not referred to.
Reply:
Thanks for the reviewer’s comments. We had referred this study in our revised discussion. In our opinion, their finding is valuable because a better collateral status would prolong the survival of penumbra, and thus correlate with larger the mismatch volume. The HIR differs from mismatch volume because the previous study found it correlates with the progression of infarct core, which make it more likely a "time" related parameter, not just "volume". That is why we would like to explore the relationship between the HIR and collaterals.
Revision in the discussion, line 290-292:
…. Better collateral circulation may extend the survival period of the penumbra. A study showed that patients with good collaterals had a smaller infarct core and higher mismatch ratio in ICA and M1/M2 occlusion and within 12 hours of stroke onset [27]. …
- A limitation of the study is that the sensitivity and specificity with this threshold are not validated in an independent dataset.
Reply:
This is indeed a major limitation of our study. We had added into our study limitation, and mentioned it in our revised discussion.
Revision in the discussion, line 321-322:
…Fifth, the sensitivity and specificity of this threshold by Syngo.via has not been validated in other independent database….
Minor comments
- Page 1. line 41-44. A reference should be provided here.
Reply:
Thanks for the reviewer’s comments. We had cited the original article bringing out the idea of HIR. [8]
Revision in the introduction, line 86-88:
Calculated by automatic software, the hypoperfusion intensity ratio (HIR) was defined as the Tmax > 10 second lesion volume (Tmax10) divided by the Tmax > 6 second lesion volume (Tmax6) [8].
- Page 1, Line 36: ‘However, there is an interrater difference in reading the result of mCTA and oijoy K. Menon et al 2015 obtaining collateral scores on mCTA. The authors have used the same protocol as in reference 6, where an excellent interrater reliability was reported: “Interrater reliability for pial arterial filling with multiphase CT angiography was excellent (n= 30, κ = 0.81, P < .001).”
Reply:
Thanks for the reviewer’s comments. We had revised the sentence to make it less contradictory. In the cited article, the 2 raters were trained radiologist. In the ideal setting without time pressure and with adequate training, we believed the reliability should be good. However, in real-world practice, the first-line practitioner who needed to interpret the result immediately may be on-duty neurologist (either visiting staff or training resident) or emergency physician, and it was always in a hurry. In our experience during daily practice and teaching course, despite not able to be quantified, there were inevitably some inter-rater differences.
Revision in the introduction, line 81-82:
…However, there may be potentially an interrater difference in reading the result of mCTA and obtaining collateral scores on mCTA in real-world setting.
- Almost same information was provided in lines (71-74) and (81-84), so these paragraphs could be merged.
Reply:
Thanks for the reviewer’s recommendation. We had revised the manuscript in both paragraphs.
Revision in the method, line 120-121:
… The patients’ demographic characteristics and medical history were recorded according to a predefined system.
Revision in the materials and methods, line 130-132:
The demographic data, last known well time or onset time, initial National Institutes of Health Stroke Scale (NIHSS) score, comorbidities, details of IV-tPA and/or EVT, and modified Rankin scale (mRS) at discharge, were obtained from our registry system.
- Line 79-80: “Patients without large vessel occlusion (LVO) or with occlusion in the anterior cerebral artery (ACA), posterior cerebral artery (PCA), posterior circulation, or multiple sites were excluded.” Can the authors explain why these patients were excluded?
Reply:
Thanks for the reviewer’s comments. The collateral score by multiphase CTA was designed for ICA and MCA occlusion. Therefore, to evaluate the correlation, we needed to exclude those without large vessel occlusion and occlusion at other sites. There is another interesting question: whether the HIR could also represent the collateral status in other stroke territory, which collaterals were difficult to evaluate. Further study may be warranted.
Revision in the materials and methods, line 129:
Patients without large vessel occlusion (LVO) or with occlusion in the anterior cerebral artery (ACA), posterior cerebral artery (PCA), posterior circulation, or multiple sites were excluded because the collateral scores were non-applicable.
- Line 90. Although the CTA protocol has been described a previous study, it would be helpful if the main CTA parameters would be described, including the timing of the different phases with respect to contrast injection.
Reply:
Thanks for the reviewer’s comments. We had added a brief description of the protocol of multiphase CTA.
Revision in the materials and methods, line 138-140:
… In brief, three phases (peak arterial phase, peak venous phase, and late venous phase) of consecutive scanning with an interval of 8 seconds were obtained, allowing for time-resolved assessment.
- Page 1 and abstract: Tmax should be defined.
Reply:
Thanks for the reviewer’s comments. We had added the definition of Tmax in the methods and cited a reference for details. However, due to the word limitation of abstract, we do not add word to define Tmax in the abstract.
Revision in the materials and methods, line 163-164:
Tmax is defined as the time to maximum of the residue function obtained by deconvolution[15].
- In the statistical analysis paragraph, they didn’t mention that they calculate inter-rater agreement using the kappa test.
Reply:
Thanks for the reviewer’s comments. We had added the calculation method in statistical analysis
Revision in the materials and methods, line 183-184:
The interrater reliability of mCTA score was measured by using Cohen’s kappa coefficient.
- A reference is missing for lines 107-109.
Reply:
Thanks for the reviewer’s comments. We had cited a reference here.
Revision in the materials and methods, line 161:
An mCTA collateral score of 3 or lower indicates poor collateral status [14].
- Twenty-six patients were excluded because of poor image quality and missing data. How the image quality assessed? Did they use an image quality score?
Reply:
Thanks for the reviewer’s comments. Those CTP with failure to be post-processed by the automatic software, or poorly enhanced mCTA, would be classified as poor image quality, usually due to severe motion artifact that couldn’t be corrected, contrast extravasation at peripheral line, poor cardiac output (e.g. heart failure with reduced ejection fraction, severe valvular heart disease) or machine/technical problem. There may not be an adequate score for image quality in these patients. We had added brief explanation in the results.
Revision in the results, line 200-201
…those with poor image quality (e.g. failure to be processed by software, poorly enhanced vessels, severe motion artifacts, etc.)
- Figure 1. There seems to be a calculation mistake in Figure 1. The differences do not match number of patients that were excluded.
Reply:
Thanks for the reviewer’s comments. We had rechecked the number again, and it seemed no mistake. Maybe our figure is misleading.
341-153=188
188-56-9-3=120
120-26=94
- Spelling mistakes in Table 1: years; Table 2: Volume (twice)
Reply:
We had corrected the spelling mistakes. Thanks a lot for the reviewer’s detailed comments and suggestions.